# Mesenchymal Stem Cells: Generalities and Clinical Significance in Feline and Canine Medicine

**DOI:** 10.3390/ani13121903

**Published:** 2023-06-07

**Authors:** Meriem Baouche, Małgorzata Ochota, Yann Locatelli, Pascal Mermillod, Wojciech Niżański

**Affiliations:** 1Department of Reproduction and Clinic of Farm Animals, Wrocław University of Environmental and Life Sciences, 50-366 Wrocław, Poland; malgorzata.ochota@upwr.edu.pl (M.O.);; 2Physiology of Reproduction and Behaviors (PRC), UMR085, INRAE, CNRS, University of Tours, 37380 Nouzilly, France; 3Museum National d’Histoire Naturelle, Réserve Zoologique de la Haute Touche, 36290 Obterre, France

**Keywords:** mesenchymal stem cells, properties, characteristics, canine, feline

## Abstract

**Simple Summary:**

Veterinary regenerative medicine is an area of active research in which mesenchymal stem cells are applied. Mesenchymal stem cells (MSCs)are cells that can be obtained from various adult tissues; these cells have an extraordinary quality of being able to self-renew and develop into other cells. MSCs can be used to treat orthopaedic conditions in dogs, asthma, kidney disease, chronic gingivostomatitis, and inflammatory bowel disease in cats. Most studies have used adipose tissue-derived MSCs because they are easily obtainable and easy to work with. However, other stem cells from different tissues may be more suitable for treating certain diseases. In this manuscript, we report the generalities and the use of mesenchymal stem cells in cats and dogs, and we believe that the ongoing research in this field will eventually bring us to a point where stem cell treatments for currently untreatable diseases will become a reality. Finally, veterinary medicine now has access to new treatments, giving hope for a cure to illnesses in our furry friends.

**Abstract:**

Mesenchymal stem cells (MSCs) are multipotent cells: they can proliferate like undifferentiated cells and have the ability to differentiate into different types of cells. A considerable amount of research focuses on the potential therapeutic benefits of MSCs, such as cell therapy or tissue regeneration, and MSCs are considered powerful tools in veterinary regenerative medicine. They are the leading type of adult stem cells in clinical trials owing to their immunosuppressive, immunomodulatory, and anti-inflammatory properties, as well as their low teratogenic risk compared with pluripotent stem cells. The present review details the current understanding of the fundamental biology of MSCs. We focus on MSCs’ properties and their characteristics with the goal of providing an overview of therapeutic innovations based on MSCs in canines and felines.

## 1. Introduction

Current cell therapies use multipotent stromal cells isolated from adult tissue, representing an emerging branch of regenerative medicine that aims to restore tissues and organs damaged by trauma, pathology, or ageing processes. Research on the therapeutic properties of stem cells in humans over many years has shown the benefits that can be obtained in inflammatory and degenerative diseases through the use of adult stem cells, particularly multipotent stromal cells or mesenchymal stem cells (MSCs) obtained from the bone marrow. Although MSC administration is performed with the support of immunosuppressive treatment, autologous MSC, which allows personalised immunomodulation, seems an interesting approach, limiting the risk of immunisation or faster apoptosis of MSC. However, this approach requires either isolation from the patients without systemic diseases or keeping the MSCs for an extended period before transplantation, causing high additional costs [1]. Several teams have turned to the use of allogeneic MSCs, allowing the creation of therapeutic batches [2]. This approach is reinforced by the fact that the injection of allogeneic MSCs seems to have the same immunoregulatory properties in vitro and in vivo. There is significant interindividual variability of the MSCs, making selecting a batch with a high immunoregulatory capacity preferable [3]. Stem cell therapy is not limited to humans. It is also of great interest in veterinary medicine and has already been used to treat animals affected by degenerative disorders, inadequate diet, and genetic disorders. It has also been used in animals with various musculoskeletal tissue injuries, primarily cartilage wear in joints and spinal discs, tendonitis, fractures, and bone degeneration [4]. Veterinarians mainly use mesenchymal stem cells (MSCs), and in recent years, treatments have been used in companion animals. MSCs provide innovative therapeutic options for diseases that previously lacked indicated treatments. Thus, protocols for regenerating damaged structures in joints, ligaments, menisci, and cartilage, similar to those observed in horses, have emerged in dogs, cats, and rabbits [5,6,7,8]. MSCs have vast potential in the treatment of many animal and human diseases. Randomised and controlled clinical studies are still necessary to apply such therapies in humans, but the success of many animal models attests to stem cells’ efficacy and therapeutic potential [9]. This review summarises the general characteristics and properties of MSCs with a particular focus on feline and canine MSCs. It also provides an overview of the use of MSCs in cell therapy and regenerative medicine.

## 2. Mesenchymal Stem Cells

MSCs are immature cells derived from mesenchyme or embryonic connective tissue, part of the mesoderm. In adults, they occur in connective tissue. MSCs are present in varying quantities and with different potentials throughout postnatal life, depending on the individual source tissue, age, and health [10,11]. The cells can be isolated significantly from different connective tissues, particularly bone marrow, umbilical cord, and adipose tissue [12]. Adult MSCs can self-renew and generate multiple types of mature and functional differentiated cells, with differentiation into specific cells of mesodermal origin (adipocytes, myoblasts, osteoblasts, and chondroblasts), depending on the environment [13,14]. However, studies have shown that MSCs can also be oriented in vitro towards endodermal phenotypes (hepatocytes, pancreatic cells) and/or ectodermal (astrocytes and epithelial cells) phenotypes [15,16]. MSCs have immunomodulation potential and positive effects on tissue tropism, and these characteristics make them ideal candidates for cell therapy and immunomodulatory strategies, particularly in systemic or local inflammatory diseases [17].

MSCs were initially isolated from the stromal compartment of bone marrow [18]. They were subsequently found in almost all postnatal connective tissues [19], umbilical cord and umbilical cord blood [20], adipose tissue [21], placental tissue [22], and cutaneous connective tissue [23]. Regardless of their course, all cells have the same basal biological characteristics, although they may differ in their potential for expansion and differentiation [24]. MSCs are a heterogeneous population of multipotent cells characterised by clonogenic abilities and differentiation potential. The International Society for Cellular Therapy’s definition of MSCs is based on three criteria: (i) their ability to adhere to plastic; a phenotype of CD73+/CD90+/CD105+ and CD45−/CD34−/CD14−, CD11b−/CD19−, or CD79a−/HLA-DR−; and their potential for differentiation into osteoblasts, chondrocytes, and adipocytes [25]. Some authors [26,27] have also suggested using other markers to select multipotent subpopulations, such as the STRO-1 marker. This marker expressed precociously on the cell surface is used to isolate mesenchymal progenitors within a cell population; its expression decreases gradually in culture. Embryonic stem cell markers, such as Oct-4, Nanog, SSEA-3, SSEA-4, TRA-1-60, and TRA-1-81, have also been described on the surface of MSCs derived from dental pulp. Some properties of MSCs are particularly promising in therapeutics, but they are only identifiable in vivo and are mainly related to the immune system.

MSCs have immunosuppressive and anti-inflammatory capabilities. The cells can modulate the immune response through their synthesis of anti-inflammatory molecules and mediators, such as interleukin (IL)-6 and macrophage colony-stimulating factor, secretion of interferon (IFN)γ, tumor necrosis factor (TNF)-α, and control of monocyte maturation [28]. MSCs have a significant role in tissue regeneration. Both transplanted and resident MSCs can contribute to tissue repair by secreting molecules involved in homeostasis, including soluble glycoproteins of the extracellular matrix, cytokines, and growth factors, which are responsible for reducing inflammation and stimulating tissue regeneration [29]. MSCs also have a significant angiogenic ability. This ability is critical for repairing and restoring organ function because oxygen supply to the tissue depends on restoring blood vessels. Some factors produced and secreted by MSCs appear to be primarily responsible for this effect. Molecules that have been identified in their secretome play a significant role in angiogenesis, including vascular endothelial growth factor (VEGF), fibroblast growth factor (FGF)-2, angiopoietin-1, chemokine (C-C motif) ligand (CCL)2, IL-6, placenta growth factor, and cysteine-rich protein and angiogenic inducer 61 [30,31]. More factors are released by MSCs, including soluble factors rich in immunomodulatory molecules, chemokines, growth factors, and cytokines. The vesicular fraction contains extracellular vesicles (EVs), which are classified primarily by their size [32,33]. Exosomes originate from the endocytic pathway and range in size from 30 to 200 nm on average and are composed of secondary metabolite, nucleic acids, proteins, and lipids. The macrovesicles originate from the cell plasma membrane and, in size from 200 to 1000 nm, contain lipids, proteins, secondary metabolites, and nucleic acids. Apoptotic bodies released by dying cells with an average between 50 and 100 μm in diameter contain nucleic acids, organelles, and proteins. All EVs participate in intercellular communication except for apoptotic bodies, which typically function in phagocytosis [32,33,34]. The properties of MSCs are maintained due to the interactions between these cells and factors in their environment, including stromal cells, signalling molecules, the extracellular matrix, and adhesion molecules. Once the cells leave these environments, they begin differentiation; however, the molecular and environmental mechanisms that control differentiation are not fully elucidated. Therefore, many studies within the veterinary field are focused on expanding the understanding of these cells [35].

## 3. The Therapeutic Role of MSCs In Vitro

The ability of MSCs to regenerate injured tissues is closely linked to their anti-inflammatory properties. MSCs act locally through cell–cell interactions based on receptor–ligand bonds or nanotubes that transfer molecules and organelles. However, they intervene mainly at the systemic level through trophic factors secreted directly in the microenvironment or transported by extracellular vesicles [36]. The MSCs can thus promote cell viability, proliferation, and angiogenesis by producing growth factors (VEGF, platelet-derived growth factor, basic fibroblast growth factor [bFGF]) [37]. They also stimulate the recruitment of endogenous stem cells by secreting chemokines, such as CXCL12 or CCL5, and reduce fibrosis by producing keratinocyte growth factor, matrix metalloproteinase (MMP)-9, bFGF, MMP-2, and hepatocyte growth factor (HGF) [38]. In addition, they intervene in the regulation of apoptosis through the production of HGF, bFGF, and insulin-like growth factor 1 (IGF1) and through the regulation of oxidative stress by releasing heme oxygenase-1 or erythropoietin. Finally, they exhibit anti-inflammatory activity by releasing indoleamine 2,3-dioxygenase (IDO), HLAG5, prostaglandin E2 (PGE2), TNFα-stimulated gene-6, IL-6, and IL-1 receptor antagonist, among other molecules. The secretion of all these factors gives these cells an unusual trophic activity [39].

## 4. Therapeutic Application of MSCs

Stem cell-based regenerative therapy is recognised as a future therapeutic option for treating many diseases in humans and animals. MSCs are good candidates for cell therapies because they are easily isolated from various tissues and have extensive and rapid proliferation [40]. The use of MSCs in regenerative medicine allows considering new therapies to treat different pathologies in cardiology, immunology, neurology, and many other diseases [41]. The development and application of cell therapy may eventually be used to treat common diseases in the population, such as diabetes and liver cirrhosis [42,43]. Cardiology could also take advantage of the advances possible through stem cell therapy. In ischemic cardiomyopathies, such as angina, acute coronary syndrome, and infarction, MSCs have shown a natural capacity to repair the heart muscle [44,45]. The transplantation of MSCs to the myocardium reduces the lesions caused by ischemia, improves wound healing, restores tissue contractile function, and increases myocardial flow by optimising left ventricular function [46,47].

Cell therapy developments are also expected to occur in neurology, with spinal cord injuries and strokes seeming to benefit from treatment with MSCs [48,49]. The injection of MSCs enhanced endogenous neuroprotection and brain plasticity through paracrine neurotrophic effects: immunomodulation, angiogenesis, synaptogenesis, oligodendrogenesis, and neurogenesis. Moreover, the apoptosis of neural cells decreased. Indeed, due to the antiapoptotic effect of certain factors, such as brain-derived neurotrophic factors, ischemic tissue was repaired, and neural function was restored [50,51]. In other investigations, clinical trials of treatments for Parkinson disease and macular degeneration have been successful owing to the ability of MSCs to increase the level of tyrosine hydroxylase and to promote the production of dopamine [52,53].

Treating neurological diseases using MSCs relies on the cells’ neuroprotective capacity after they migrate into damaged brain tissue. Although MSCs benefit rain lesions and tissue through various trophic factors, such as nerve growth factor (NGF), brain-derived neurotrophic glial-derived neurotrophic factor (GDNF), vascular endothelial growth factor (VEGF), and insulin-like growth factor (IGF), they also have immunomodulatory, angiogenic, and antiapoptotic effects [54,55]. In addition, they stimulate endogenous regeneration by activating neural progenitor cells quiescent in brain tissue [56,57]. Other medical fields use MSCs to influence wound healing, angiogenesis, and reepithelialisation [58]. They also seem to regenerate the function of specific specialised tissues, such as sweat glands [59]. In addition, clinical trials on ischemic tissue regeneration in diabetic patients have shown revascularisation and healing of damaged tissue after stem cell treatment [60].

The immunomodulatory properties of MSCs arouse great clinical interest. Their ability to produce trophic, immunomodulatory, and immunosuppressive factors enables their use to treat graft-versus-host and certain autoimmune diseases [61,62]. In addition, the application of MSCs in treating type 1 diabetes results in the arrested destruction of β cells, increasing the differentiation of stem cells into insulin-producing cells and tissue repair by stabilising the inflammatory response [63]. However, using MSCs in cartilage regeneration is still challenging because large amounts of cells need to be injected [64]. Recently, MSCs were used to treat premature ovarian insufficiency. The results were not precise, but the procedure offers a promising treatment option to improve lipid metabolism and restore ovarian function by activating the phosphoinositide 3-kinase pathway, promoting the level of free amino acids, and reducing the concentration of monosaccharides [65]. MSCs-derived secretomes have been used in different clinical trials and shown to produce the same or even enhanced therapeutic effect compared with MSCs [66]. Moreover, MSC-derived secretomes have been shown to display a dual function in tumor promotion and tumor suppression [67].

## 5. Veterinary Use of Mscs in Companion Animals

### 5.1. Canine MSCs

#### 5.1.1. Sources and Characteristics

Canine MSCs (cMSCs) were initially obtained from adipose tissue [68]. They have since been isolated from allogenic and autologous sources, including bone marrow muscle and periosteum [69], umbilical cord blood [70], Wharton’s jelly [71], umbilical cord tissue [72] amniotic membrane [73], amniotic fluid [74], the limbal epithelium [75], endometrium [76], and the dental pulp [77]. In addition, cMSCs have also been harvested from olfactory epithelium [78], periodontal ligament [79], synovium [80], placenta [81], peripheral blood [82], and ovary [83]. cMSCs obtained from different sources are plastic adherent with a spindle-shaped morphology. Some studies have shown that the morphology of cMSC varies from more cuboid to very thin with cytoplasmic extension [84]. The cells are positive for CD90, CD105, CD44, and CD73 markers, but they lack the expression of the hematopoietic cell surface markers CD34, CD45, CD146, and CD11b. In addition, some studies have shown the absence of other markers, such as CD14, D11b, CD19, CD29α, CD45, CD34, and HLA-DR [85]. The different sources and genetic differences between various breeds may cause variation in their biological characteristics; dissimilarities in multi-lineage differentiation and proliferation level can define their clinical uses [86].

#### 5.1.2. Canine MSC Therapy

Dogs have been extensively studied in cartilage repair work because, like humans, dogs lack an intrinsic repairability of cartilage and can experience the same cartilage diseases as humans, including osteoarthritis and osteochondritis dissecans [87,88].

##### Osteoarthritis (OA)

Canine osteoarthritis (OA) is a degenerative disease of joint tissues that leads to the loss of joint cartilage and the release of inflammatory and regulating cytokines, causing pain. The cartilage’s ability to heal is inadequate because of its avascular nature. After a lesion, the fibrous tissue is formed with various functional properties of the native hyaline cartilage, promoting joint degeneration [89,90]. The OA pathophysiology is multifactorial, with a robust inflammatory component, and it is frequently secondary to anatomical anomalies or injuries, causing joint instability. It is widespread in large animals but can also affect dogs from all breeds; there is no cure for OA, and the treatment routine focuses on pain reduction and symptom management [89,91]. Conventional treatment is based on diet, long-term nonsteroidal anti-inflammatory drugs, weight management, and dietary supplements. More therapies have been used and studied, such as acupuncture and shockwave therapy. More recently, MSCs have been used as a promising tool for treating different OA cases [91,92]. Studies evaluated the therapy improvement of hip joint OA using the subjective method, including a range of motion scores, pain, and lameness [93,94]. Black et al. [94] conducted a study on 21 dogs with chronic hip joint OA: the dogs were treated with 4.2–5 × 10^6^ intra-articular autologous ASCs for 6 months, and the study results showed a significantly improved score for pain, lameness, and range of motion compared with the control group. Marx et al. [93] evaluated the effect of allogeneic ASCs, and autologous stromal vascular fraction injected into acupuncture in 6 dogs; after 60 days, all 5 dogs showed an improvement in lameness, range of motion, and pain manipulation. More studies carried out by Viral et al. [95,96] used the objective method to analyse the approach using a force platform to demonstrate the effectiveness of a single AI injection of ASCs. The first research revealed how the effect of the combination of ASCs and PRGF was extended over 6 months [95]. In the second study, the same team showed that using ASCs alone improves the dogs’ conditions after the first month of the treatment with a reduction in lameness and pain; however, this effect gradually eased between the first and the third month [96]. In the third study, Vilar and his team [97] used the force platform to compare the pain scales for the same animal treated with ASCs six months after therapy. The results showed that using pain assessment scales to measure lameness associated with OA did not reveal high accuracy compared with the quantitative force platform gait approach [97].

Numerous studies have shown notable results using the intra-articular administration of ASCs for canine elbow OA therapy, with improved pain, lameness, amplitude of motion, and functional capacity [93,98,99]. Éva Kriston-Pál et al. [98] used MSCs resuspended in 0.5% hyaluronic acid to treat dogs suffering from elbow dysplasia and OA; the results reported a significant improvement demonstrated by the degree of lameness during the follow-up period of one year. Controlled arthroscopy also showed that cartilage had completely regenerated in one dog. In a more recent study, Olsen et al. [99] used IV injections of allogeneic ASCs (1–2 × 10^6^ cells/kg body weight) to treat 13 dogs with elbow OA 2 weeks apart. No acute adverse effects were observed, and a significant improvement in clinical signs and the owner’s perception was noted. However, synovial fluid OA biomarkers did not change after MSCs administration. Despite subjective outcomes showing good enhancements, such as the dog’s clinical signs, objective outcome measures did not confirm similar results, such as reducing the OA biomarkers measurement in synovial fluid. Larger sample sizes and CGs are needed to interpret these findings [99]. According to previous studies, treating OA using MSCs in combination or alone improves the clinical signs, reducing lameness and providing remarkable recovery after the last limited sports activity [100,101,102,103].

##### Osteochondritis

Dissecting osteochondritis is common in large dogs of predisposed breeds before the age of one year. The joints mainly affected are the shoulders, elbows, and ankles (tarsus). The treatment uses arthroscopically guided excision of free cartilage fragments in the joint combined with stem cells and plasma enriched with growth factors [101]. Robert Harman et al. (2016) carried out a study of a treatment for osteoarthritis using MSCs obtained from adipose tissue. In this trial, 43 dogs in the treatment group received a dose of 12 × 10^6^ cryopreserved allogeneic MSCs intra-articularly. The study measured the effects of MSCs on pain during handling and assessed the dogs’ abilities to perform daily activities for two months [90]. No severe side effects were associated with the treatment in this study, and there was a notable reduction in pain and improved functional abilities. Intra-articular injection of MSCs has also proved to be a promising technique [102].

##### Tendonitis and Ligaments Rupture

In dogs, tendonitis is another frequently diagnosed disorder that can cause significant locomotor disorders. In addition to tendonitis, ruptures and lacerations are other common tendon disorders in dogs [103]. These disorders rarely resolve spontaneously and invariably require treatment followed by physiotherapy; therefore, an effective treatment that heals the scar tissue as closely as possible to resemble the healthy tendon properties is needed. Autologous adipose MSCs used in the tendon treatment modulate the tendon’s post-repair inflammatory response by increasing prostaglandin reductase1, M2 macrophage, and proteins involved in tendon formation. Moreover, the anti-inflammatory effect of MSCs is thought to cause a decrease in collagen fibre alteration [95,104]. Currently, MSC therapy is an exciting prospect. Studies have demonstrated a histologically significant improvement in tendon healing following treatment with adipose-derived progenitor cells (ADPC) or bone marrow aspirate concentrate (BMAC) and platelet-rich plasma (PRP) combination on partial cranial cruciate ligament rupture CCL. This investigation reviewed 36 medical records of client-owned dogs diagnosed with an early partial tear of the craniomedial band of the CCL treated with BMAC–PRP or ADPC–PRP from 2010 to 2015. The data collected are mainly the results of the diagnostic arthroscopy on days 0 and 90, the physical and orthopaedic examination, the medical history, the x-rays, and the objective analysis of the temporospatial gait [105]. In another study, dogs with unilateral cranial cruciate ligament rupture confirmed by arthroscopy were treated as follows: The first group received an intra-articular injection of allogeneic neonatal MSCs after tibial plateau levelling osteotomy, followed by a placebo for one month. The second group received the same concentration of MSCs after tibial plateau levelling osteotomy, followed by nonsteroidal anti-inflammatory drugs (NSAIDs). After one month, the results showed tendon healing in the group treated with MSCs. The same result was recorded in the other group treated with nonsteroidal anti-inflammatory drugs, and insignificant differences between the two groups in gait evaluation after three months were reported [106].

Other studies have explored the use of MSCs to treat systemic or local inflammatory pathologies and autoimmune diseases in dogs [107]. In addition, the canine model offers certain advantages, such as the possibility of conducting long studies involving physiotherapy or exercise protocols. Finally, dogs are considered model animals for human research [108].

### 5.2. Feline MSCs

#### 5.2.1. Sources and Characteristics

MSCs have been isolated from different tissues in cats. The initial isolation of MSCs from bone marrow and characterisation of the cells were reported in 2002, followed by isolation from fetal fluid, fat, peripheral blood, amniotic membrane, umbilical cord blood [7], and from different parts of the umbilical cord tissue [109]. As with all MSCs, feline MSCs (fMSCs) have the capacity for self-renewal. They also display a typical fibroblast-like appearance and plastic adherence, express numerous surface markers (CD90, CD44, CD105), and are negative for leukocyte markers (CD4, CD18) and histocompatibility complex (MHC) II [110]. These characteristics can be altered after extended culture. Lee et al. [111] showed that proliferation and the expression of surface markers of adipose-derived fMSCs decreased after multiple passages; for this reason, their use in cell therapy will be more effective during the early passages. fMSCs also have the potential to differentiate into adipogenic, osteogenic, and chondrogenic cells [112].

MSCs can modulate both adaptive and innate immune systems: T lymphocytes are the primary mediators of the adaptive immune response, and fMSCs have the same immunomodulatory gene expression and response to inflammatory cytokines as human MSCs. The secretion of IFN-γ and TNF-α stimulate MSCs, which attracts T lymphocytes by chemotaxis for cell contact; however, MSCs are poor immunogenic cells because they do not express HLA class II molecules HLA-DR or the costimulatory molecules CD40, CD80, and CD86 [113,114,115]. As a result, they escape recognition by CD4+ T lymphocytes and cause them to become energy sources [115]. As a result, they escape recognition by CD4+ T lymphocytes and cause them to become energy sources [116]. These molecules inhibit the proliferation of T lymphocytes and the activation of T lymphocytes by antigen-presenting cells, and they induce the differentiation and survival of regulatory T lymphocytes. In addition, the expression of IDO enzyme by fMSCs inhibits the proliferation of T lymphocytes by reducing the amount of tryptophan in the surrounding environment, an amino acid essential for cell multiplication [117].

In later passages, FMSCs develop giant foamy multinucleated cells, causing proliferation arrest and syncytial cell formation. These cytopathic effects are caused by infection with the feline foamy virus (FeFV), a very common, asymptomatic retrovirus in cats. The impacts of FeFV infection on fMSC function make their use in therapy impossible [118,119,120,121,122,123,124]. However, a recent study conducted by Boaz et al. shows that treating fMSCs infected by FeFV using an antiretroviral drug, tenofovir, in early passages effectively prevents the harmful effects of the infection and supports in vitro expansion [119].

Current and potential clinical applications of mesenchymal stem cell therapy in cats are explored in several diseases. Based on the immunomodulatory properties of feline stem cells, clinical trials show interest in this new therapeutic strategy for treating illnesses such as gingivostomatitis, chronic inflammatory bowel diseases, asthma, and even kidney failure [120].

#### 5.2.2. Feline MSCs and Their Clinical Use

Current and potential clinical applications of MSC therapy in cats have been investigated in several diseases. Based on the immunomodulatory properties of feline stem cells, clinical trials have been conducted for the treatment of disorders such as gingivostomatitis, chronic inflammatory bowel diseases, asthma, and even kidney failure [121].

##### Feline Asthma

Cats with asthma have a progressive decline in respiratory function linked to structural remodelling of the airways, characterised by subepithelial fibrosis and bronchial smooth muscle hypertrophy. These structural changes result from communication between cells of different bronchial structures, including fibroblasts, epithelial cells, smooth muscle cells, and immune cells present within the bronchial mucosa [122]. Current therapies for asthma are mainly based on steroidal anti-inflammatory drugs, but many side effects appear over time. MSCs have been used to treat cats with chronic, acute, and allergic asthma. Cats have been treated in different ways; in one study, adipose-derived MSCs intravenous injections at a dose (0.36–2.5 × 10^7^ MSCs/infusion) were administered every two months over one year. Results show that MSCs positively affect airway remodelling at eight months, diminished airway hyperresponsiveness, and decreased airway eosinophilia compared with placebo, but no effect on airway inflammation [123]. In the second study, serial intravenous infusions of allogeneic, adipose-derived MSCs were administered at different doses (2 × 10^6^, 4 × 10^6^, 4.7 × 10^6^, 1 × 10^7^, and 1 × 10^7^); after 133 days, treatment of allergic asthma experimentally induced by allogeneic MSCs resulted in significant improvement in airway hyperresponsiveness, airway inflammation, and airway remodelling [124]. These two studies showed that MSCs positively affected airway remodelling, diminished airway hyperresponsiveness, and decreased airway eosinophilia compared with placebo but had no effect on airway inflammation and improvement [123,124].

##### Feline Kidney Disease

The effectiveness and safety of MSC administration were investigated in the treatment of cats with chronic kidney disease (CKD), using a single unilateral intrarenal injection of autologous adipose tissue-derived or bone marrow-derived MSC (bmBM MSC or aMSC) via ultrasound guidance. A total of 6 cats were used for this study, including 2 healthy 1.5-year-old cats and 4 cats with CKD whose ages varied between 6 and 13 years. Intrarenal injection resulted in a mild decrease in serum creatinine concentration and a modest improvement in glomerular filtration rate without inducing adverse effects [125,126,127,128,129,130,131]. In another investigation, cats with naturally occurring CKD were treated with feline amniotic membrane-derived allogenic MSCs via internal and intravenous injection [127]. Unfortunately, the internal injection of aMSCs was unsuccessful because of stress, sedation, bleeding, and anaesthesia complications. In contrast, data obtained from intravenous administration revealed significant improvement in proteinuria, decreased serum creatinine, and mild improvements in urine-specific gravity [128].

In another research study, the treatment of acute kidney injury (AKI) in an ischemic kidney model in adult research, cats underwent unilateral renal ischemia for 60 min with fibroblasts (five cats), aMSCs (five cats), or bm-MSCs (five cats). Three cats that had undergone ischemia previously were used as a control. The results of the study revealed no AKI influence or smooth muscle actin staining [129]. Thus, despite decreased serum creatinine concentrations, using MSCs in treating CKD did not lead to a clinically meaningful improvement in renal function. Furthermore, none of the tests in cats with CKD have reproduced the positive results obtained in rodents [129].

##### Feline Chronic Gingivostomatitis

Feline chronic gingivostomatitis (FCGS) results from an inadequate immune response of the cat to different antigenic stimulations. The disorder affects the gums and other parts of the oral cavity, and its treatment is long and complex [130]. Nevertheless, the ability of MSCs to downregulate the activation of T lymphocytes makes their use in the treatment of chronic stomatitis in feline medicine remarkable [131] Boaz Arzi et al. applied allogenic aMSCs in clinical cases of FCGS: FCGS cats refractory to full-mouth tooth extraction were enrolled [132,133]. In each trial, 7 FCGS cats received 2 intravenous injections of 2 × 10^7^ aMSCs 3–4 weeks apart. In the first experiment, the seven cats received autologous aMSCs; in the second experiment, the seven cats received unmatched, allogeneic aMSCs from SPF donor cats. The results demonstrated that cats treated with allogenic and autologous aMSCs recovered from their clinical condition, with the clinical cure being shown by the histopathology resolution of B- and T-cell inflammation. Moreover, neutrophil counts, normalisation of the CD4/CD8 ratio, and numbers of circulating CD8+ T cells were decreased, while serum IL-6 and TNF-α concentrations were temporarily increased [132,133,134,135,136,137,138].

In a different experiment, allogenic aMSCs were used to treat cats with chronic clinical enteropathy, and the results showed no side effects, with a significant improvement in clinical signs [132].

##### Inflammatory Bowel Disease

Feline inflammatory bowel disease (IBD) is a condition in which a cat’s gastrointestinal (GI) tract becomes chronically irritated and inflamed; the possible causes can include bacterial or parasitic infection, intolerance, or allergy to a specific protein in the diet [7,134]. To date, there is no single best treatment for IBD, so veterinarians may need to try several combinations of medications or diet to determine the best therapy [135]. For this reason, alternative approaches became necessary; applying MSCs as an alternative treatment for IBD is still a very recent concept in veterinary medicine [120]. Tracy et al. [136] conducted a clinical trial involving seven cats with diarrhea for at least three months. They received two IV injections of 2 × 10^6^ cells/kg from cryopreserved feline ASCs, while four cats with a similar clinical condition received a saline placebo. Clinical signs improved in five out of seven cats treated with stem cells after one to two months, unlike the placebo group, which did not show any progress. With this trial, it is possible to conclude that MSC therapy was well tolerated and potentially effective in treating feline chronic enteropathy. However, these preliminary results require a significant follow-up study.

In recent research performed by Tracy et al. [137] fMSCs were used as a treatment for IBD after the failed diet trial and compared with prednisolone treatment. The endoscopic biopsies confirmed the histopathologic diagnosis of IBD, and the cats were randomly assigned to either the prednisolone or fMSC groups.

In total, 12 cats were treated, 6 cats in each group. The cats that received fMSCs were between 4.5 to 13 years old and included 3 neutered males and 3 spayed females with weights varying between 4 and 5.9 kg; cats received IV injections of 2 × 10^6^ cells/kg of freshly allogenic adipose-derived MSCs separated by 2 weeks. The prednisolone group included spayed females with a mean age of 8.3 years and a mean weight of 3.6 kg. They received a 1–2 mg/kg PO q24h. In each group, one cat failed the treatment at the second-month recheck, and five completed the six months study with no changes in diet and medications [137]. The results showed that freshly allogenic adipose-derived MSCs were safe and easily administrated in the cat with IBD without any side effects; the response to therapy was similar between the group that received MSC infusions and the group that received standard prednisolone therapy. However, a more extensive study is needed to confirm the efficacy and duration of the effect [137].

The studies were carried out on a small number of treated animals, making the published results interesting and promising. More studies would certainly be required to confirm its beneficial influence. Other diseases that could benefit from this new therapeutic strategy in veterinary medicine remain to be investigated.

## 6. Conclusions

Along with a scientific interest in regenerative medicine, interest in MSCs has grown over time. Many studies have made it possible to characterise these cells and demonstrate their regenerative potential, and it appears that their use in new therapeutic approaches is inevitable. Indeed, two essential properties of MSCs make them critical in regenerative medicine: their ability to proliferate without losing their undifferentiated character and ability to differentiate into specialised cells. Other properties found more recently, such as their abilities to modulate the immune system and to secrete molecules influencing their environment, make them even more attractive. The evolution of our understanding of MSCs and their use will enable the development of new therapeutic strategies, particularly in veterinary regenerative medicine. In addition, MSC therapy is a promising option for treating several diseases.

Nevertheless, many factors remain to be investigated regarding the protocols of use, the most suitable source of stem cells, the optimal route of administration, and the impact of the donor’s status on stem cell function. For that reason, when selecting a donor for cell-based products in veterinary clinical trials, screening them for infectious diseases and other risk factors is crucial to prevent the transmission of disease agents and ensure the safety of the animal subjects involved in the trial. Therefore, besides the quality-controlled cells, it is essential to clearly understand their origin, storage conditions, and product composition. In addition, it is also necessary to demonstrate that cellular function and integrity have been preserved throughout the process and prove that the cells are free of contamination from viruses, bacteria, fungi, mycoplasma, and endotoxins.

Conducting long-term safety evaluations to ensure no adverse effects is highly recommended. If any adverse events occur after stem cell intervention, reporting them and, more importantly, considering the potential risk factors, such as toxicity, tumorigenicity, and immune reactions, is essential. Moreover, there are regulations and guidelines for using stem cell-based products in veterinary practice made by the European Medicine Agency (EMA), the United States Food and Drug Administration (FDA), and the Animal and Plant Quarantine Agency (APQA) of Korea [138,139,140,141] to ensure the safety assessment of cell-based products for animal use.

## Data Availability

Not applicable.

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
