# Peer review of "Mesenchymal Stem Cells: Generalities and Clinical Significance in Feline and Canine Medicine"

_animals, 2023, doi:10.3390/ani13121903_

Round 1

Reviewer 1 Report

1. Check the abbreviation in "5. Veterinary Use of Mscs in Companion Animals".

Author Response

Check the abbreviation in "5. Veterinary Use of MSCs in Companion Animals".

The authors are grateful for the valuable comment and good evaluation.  

The abbreviations in section 5, Veterinary Use of MSCs in companion Animals, were checked, and the whole nomination was added in the manuscript as follows: Although MSCs benefit rain lesions and tissue through various trophic factors such as nerve growth factor (NGF), glial-derived neurotrophic factor (GDNF), Vascular endothelial growth factor (VEGF), and Insulin-like growth factor (IGF).

Reviewer 2 Report

In their present paper M. Baouche et al. give  a comprehensive overview of mesenchymal stem cells (MSCs) and their potential applications and significance in feline and canine medicine.

The manuscript is well written and generally informative, although the number of references is extensive and could be considerably shortened, focusing on the real important achievements. While the review highlights the many potential benefits of MSCs, it is also important to note that much more basic research is required before these therapies can be widely implemented. It would be good, if the role of MSCs during embryo normal embryo development in cats and dogs is shortly addressed.

This interesting review shows that one of the main challenges associated with using MSCs in veterinary medicine is the lack of standardization in terms of cell isolation, culture, and characterization. This makes it difficult to compare results across studies and to determine which protocols are most effective. As result of their review the authors should give some suggestions, how these problems can be overcome in the future. Another challenge is related to safety and efficacy. While MSCs have shown promise in treating a wide range of conditions, there are still many unanswered questions about their long-term effects and potential risks. For example, there is some concern that MSCs could promote tumor growth or cause other adverse effects. It would be good, if the results of human studies should be compared to a greater extent with the data in dogs and cats, considering the different life expectancy of these species.

In conclusion, after a thorough answering these points and correction of some spelling errors I would recommend this paper for publication in Animals

The manuscript is well written and only the correction of some minor spelling errors is necessary.

Author Response

I would like to take this opportunity to express our thanks to the Reviewer for the positive feedback and helpful comments for correction or modification. My co-authors and I very much appreciated the encouraging, critical and constructive comments on this manuscript by the Reviewer. The comments have been very thorough and useful in improving the manuscript. All suggested changes have been addressed as follows:

In their present paper, M. Baouche et al. give a comprehensive overview of mesenchymal stem cells (MSCs) and their potential applications and significance in feline and canine medicine.

The manuscript is well written and generally informative, although the number of references is extensive and could be considerably shortened, focusing on the real important achievements. While the review highlights the many potential benefits of MSCs, it is also important to note that much more basic research is required before these therapies can be widely implemented. It would be good if the role of MSCs during embryo normal embryo development in cats and dogs is shortly addressed.

The authors are grateful for this comment. Unfortunately, we reviewed most of the published data thoroughly to gain a comprehensive understanding of the topic at hand which made the number of references extensive, authors tried to shorten it, but as the other revivers requested some additions to the manuscript, we were not able to comply with this request.

Authors agree with the Reviewer that it will be good to add the role of MSCs during embryo development in cats and dogs but from the literature data, MSCs have been mainly utilised as a coculture in supporting oocyte maturation and embryo development in pigs, mice, and cattle among various animal species, which is not within the scope of this review.

Additionally, researchers have used cumulus cells, oviduct cells, and extracellular vesicles to improve oocyte maturation in dogs, but not MSCs cells. Concerning cats, no such data has been found in the available literature.

This interesting review shows that one of the main challenges associated with using MSCs in veterinary medicine is the lack of standardization in terms of cell isolation, culture, and characterization. This makes it difficult to compare results across studies and to determine which protocols are most effective. As a result of their review, the authors should give some suggestions on how these problems can be overcome in the future.

The following paragraph was added at the end of the conclusion as a result of the review to show how the problem of standardization can be overcome in the future.

Lines 476-483: For that reason, when selecting a donor for cell-based products in veterinary clinical trials, screening them for infectious diseases and other risk factors is crucial to prevent the transmission of disease agents and ensure the safety of the animal subjects involved in the trial. Therefore, besides the quality-controlled cells, it is essential to understand their origin, storage conditions clearly, and product composition. It is also necessary to demonstrate that cellular function and integrity have been preserved throughout the process and prove that the cells are free of contamination from viruses, bacteria, fungi, mycoplasma, and endotoxins.

Another challenge is related to safety and efficacy. While MSCs have shown promise in treating a wide range of conditions, many unanswered questions remain about their long-term effects and potential risks. For example, there is some concern that MSCs could promote tumor growth or cause other adverse effects. It would be good if the results of human studies should be compared to a greater extent with the data in dogs and cats, considering the different life expectancies of these species.

The authors fully agree with this comment of the reviewer, but we were  asked to reduce the use of the content on animal models and human medicine in our manuscript as animals journal did not publish that kind of article; for that, we tried to add only short sentences mentioning the safety and the efficacity of MSCs in companion animals at the end of conclusion, as follow:

Lines 484-491: Conducting long-term safety evaluations to ensure no adverse effects is highly recommended. If any adverse events occur after stem cell intervention, reporting them and, more importantly, considering the potential risk factors, such as toxicity, tumorigenicity, and immune reactions is essential. Moreover, there are regulations and guidelines for using stem cell-based products for veterinary made by the European Medicine Agency (EMA), the United States Food and Drug Administration (FDA) and the Animal and Plant Quarantine Agency products for animal use.

The following references have been added to the manuscript.

  1. Stem Cell-Based Products for Veterinary Use: Specific Questions on Target Animal Safety to Be Addressed by ADVENT [Internet] London: European Medicines Agency; [Updated 2016]. [Accessed 2018 Sep 12]. http://www.ema.europa.eu/docs/en_GB/document_library/Scientific_guideline/2016/07/WC500210915.pdf.
  2. Cell-Based Products for Animal Use [Internet] Rockville: Food and Drug Administration; [Updated 2015]. [Accessed 2018 Sep 12]. https://www.fda.gov/downloads/AnimalVeterinary/GuidanceComplianceEnforcement/GuidanceforIndustry/UCM405679.pdf.
  3. Questions and Answers on Allogenic Stem Cell-Based Products for Veterinary Use: Specific Questions on Sterility [Internet] London: European Medicines Agency; [Updated 2017]. [Accessed 2018 Sep 12]. http://www.ema.europa.eu/docs/en_GB/document_library/Scientific_guideline/2017/06/WC500229927.pdf.
  4. Guideline on Safety Assessment of Cell-Based Products for Animal Use [Internet] Gimcheon: Animal and Plant Quarantine Agency.

Reviewer 3 Report

I think the ms is a well written paper; just a minor revision of the english should be considered.

A minor revision of the english should be considered.

Author Response

The authors are grateful for the valuable comment and good evaluation

I think the ms is a well-written paper; just a minor revision of the English should be considered.

A minor revision of the English should be considered.

The article has been proofread and corrected by a native English speaker.

Reviewer 4 Report

Very interesting article, however, the composition of the secretome from each cell line could also be explored, trying to understand and characterize each component that may or may not have a greater impact on the regeneration of each tissue.

It would also be interesting to refer to studies that exist comparing the use of the cellular system versus just its secretome.

More bibliographic references could have been presented on the subject

Author Response

The authors are very grateful for the valuable comments that help to improve our manuscript. We very much appreciate the encouraging, critical, and constructive comments on this manuscript. The comments have been very thorough and useful in improving the manuscript. All suggested changes have been addressed as follows:

Very interesting article, however, the composition of the secretome from each cell line could also be explored, trying to understand and characterise each component that may or may not have a greater impact on the regeneration of each tissue.

The authors fully agree with the reviewer and MSCs secretome was added briefly to the manuscript in section: 2. Mesenchymal stem cells.

Lines118-127: factors produced and secreted by MSCs appear to be primarily responsible for this effect. The soluble factors are rich in Immunomodulatory molecules, chemokines, growth factors and cytokines. The vesicular fraction contains extracellular vesicles (EVs), which are classified primarily by their size. Exosomes originate from the endocytic pathway and range in size from 30 to 200 nm on average and are composed of secondary metabolite, nucleic acids, proteins, and lipids[30,31]. The microvesicles originate from the cell plasma membrane and, in size from 200 to 1000 nm, contain lipids, proteins, secondary metabolites, and nucleic acids. Apoptotic bodies released by dying cells with an average between 50 and 100 μm in diameter contain nucleic acids, organelles, and proteins. All EVs participate in intercellular communication, with the exception of apoptotic bodies, which typically function in phagocytosis.

It would also be interesting to refer to studies that exist comparing the use of the cellular system versus just its secretome.

The use of MSCs secretome was added to the manuscript in section 3 to shorten the way.

Lines: MSCs-secretomes were used in different clinical trials and shown to produce the same therapeutic effect or are even enhanced in comparison to MSCs. Moreover, MSC-derived secretomes have been shown to display a dual function in tumor promotion and tumor suppression.

More bibliographic references have been added to the manuscript.

  1. Suire, C.N.; Hade, M.D. Extracellular Vesicles in Type 1 Diabetes: A Versatile Tool. Bioengineering 2022, 9, 105, doi:10.3390/bioengineering9030105.
  2. Wang, X.; Zhang, Z.; Yao, C. Angiogenic Activity of Mesenchymal Stem Cells in Multiple Myeloma. Cancer Invest 2011, 29, 37–41, doi:10.3109/07357907.2010.496758.
  3. Hofer, H.R.; Tuan, R.S. Secreted Trophic Factors of Mesenchymal Stem Cells Support Neurovascular and Musculoskeletal Therapies. Stem Cell Res Ther 2016, 7, 131, doi:10.1186/s13287-016-0394-0.
  4. Műzes, G.; Sipos, F. Mesenchymal Stem Cell-Derived Secretome: A Potential Therapeutic Option for Autoimmune and Immune-Mediated Inflammatory Diseases. Cells 2022, 11, 2300, doi:10.3390/cells11152300.
  5. Muralikumar, M.; Manoj Jain, S.; Ganesan, H.; Duttaroy, A.K.; Pathak, S.; Banerjee, A. Current Understanding of the Mesenchymal Stem Cell-Derived Exosomes in Cancer and Aging. Biotechnology Reports 2021, 31, e00658, doi:10.1016/j.btre.2021.e00658.